# Protection of Citrus Fruits from Postharvest Infection with *Penicillium digitatum* and Degradation of Patulin by Biocontrol Yeast *Clavispora lusitaniae* 146

**DOI:** 10.3390/microorganisms8101477

**Published:** 2020-09-25

**Authors:** Mariana Andrea Díaz, Martina María Pereyra, Fabricio Fabián Soliz Santander, María Florencia Perez, Josefina María Córdoba, Mohammad Alhussein, Petr Karlovsky, Julián Rafael Dib

**Affiliations:** 1Planta Piloto de Procesos Industriales Microbiológicos (PROIMI) - Consejo Nacional de Investigaciones Científicas y Técnicas (CONICET), Av. Belgrano y Pje. Caseros, 4000 Tucumán, Argentina; dmarianaandrea@gmail.com (M.A.D.); martinapereyra30@gmail.com (M.M.P.); fabianfabriciosoliz@gmail.com (F.F.S.S.); mfp_1206@hotmail.com (M.F.P.); cordobajosefina@hotmail.com (J.M.C.); 2Molecular Phytopathology and Mycotoxin Research, University of Goettingen, Grisebachstrasse 6, D-37077 Göttingen, Germany; malhuss@uni-goettingen.de; 3Instituto de Microbiología, Facultad de Bioquímica, Química y Farmacia, Universidad Nacional de Tucumán, Ayacucho 471, 4000 Tucumán, Argentina

**Keywords:** patulin, postharvest disease, *Penicillium*, *Clavispora lusitaniae*, sensorial analysis, biocontrol spectrum, citrus

## Abstract

Fungal rots are one of the main causes of large economic losses and deterioration in the quality and nutrient composition of fruits during the postharvest stage. The yeast *Clavispora lusitaniae* 146 has previously been shown to efficiently protect lemons from green mold caused by *Penicillium digitatum*. In this work, the effect of yeast concentration and exposure time on biocontrol efficiency was assessed; the protection of various citrus fruits against *P. digitatum* by *C. lusitaniae* 146 was evaluated; the ability of strain 146 to degrade mycotoxin patulin was tested; and the effect of the treatment on the sensory properties of fruits was determined. An efficient protection of lemons was achieved after minimum exposure to a relatively low yeast cell concentration. Apart from lemons, the yeast prevented green mold in grapefruits, mandarins, oranges, and tangerines, implying that it can be used as a broad-range biocontrol agent in citrus. The ability to degrade patulin indicated that strain 146 may be suitable for the control of further *Penicillium* species. Yeast treatment did not alter the sensory perception of the aroma of fruits. These results corroborate the potential of *C. lusitaniae* 146 for the control of postharvest diseases of citrus fruits and indicate its suitability for industrial-scale fruit processing.

## 1. Introduction

Fungal diseases are one of the main causes of large economic losses and deterioration in the quality and nutrient composition of fruits during the postharvest stage. They contribute significantly to the reduction of the shelf life of products during storage, contaminate fruits with mycotoxins, and reduce their market value. In the fresh production supply chain, such drawbacks have traditionally been overcome through the use of synthetic chemicals. However, due to concerns that intensive and sustained use of fungicides negatively affects humans and the environment, biological alternatives to chemical postharvest disease control are being developed [1,2,3,4,5,6]. Biological control of postharvest spoilage has attracted attention particularly in citrus fruits [7,8].

The most aggressive postharvest fungal pathogens of lemons and other fruits belong to the genus *Penicillium* [9,10]. Several *Penicillium* pathogens, notably *Penicillium expansum* and *Penicillium griseofulvum*, produce the mycotoxin patulin [11], posing a concern for food safety [12,13,14]. The major postharvest pathogen of citrus fruits is *Penicillium digitatum*, the cause of green mold [10]. In recent years, the biology of *P. digitatum* and virulence factors involved in the colonization of fruits by this pathogen have been studied on biochemical, transcriptome, and molecular levels [15,16,17,18]. At the same time, new options for controlling *P. digitatum* in fruits have been investigated. Although new chemical agents [19,20,21] and peptides [22] for the control of *P. digitatum* are being developed, biological control remains the most promising option [8]. Among biocontrol agents, antagonistic yeasts have been the most frequently tested organisms [3,8,10,23,24,25,26,27,28,29]. Recent work highlighted the use of the killer yeast *Clavispora lusitaniae* strain 146 as a promising biocontrol agent against *P. digitatum*. Besides its preventive activity against green mold in lemons, *C. lusitaniae* 146 showed a high tolerance to fungicides normally used in citrus packinghouses, suggesting a possible combined use in order to reduce or avoid the utilization of synthetic fungicides [27]. Moreover, strain 146 has the ability to maintain protection during an entire harvest period of lemons along with the ability to colonize wounds in lemons at both low and room temperatures [27,28].

Based on these encouraging results, this work focuses on expanding our knowledge about the biocontrol activity of *C. lusitaniae* 146. The impact of different factors on the biocontrol effect was studied, such as yeast growth phase, cell concentration, and dipping time. Since yeast produces aromatic compounds that could negatively influence product quality [30], the consumer preference between yeast-treated and non-treated lemons was evaluated. Some microorganisms used in the biocontrol of plant pathogens not only inhibit fungal growth but also reduce mycotoxin levels [29,31,32,33]. From the perspective of extending the application of *C. lusitaniae* 146 to further *Penicillium* species, some of which produce patulin, the ability of the yeast to degrade patulin was assessed, using the yeast *Rhodosporidium paludigenum* as a reference strain [34]. Finally, the biocontrol efficiency of *C. lusitaniae* 146 was evaluated in citrus fruits other than lemons.

## 2. Materials and Methods

### 2.1. Microorganisms, Culture Conditions, and Fruits

The killer yeast used in this study was previously isolated and characterized as described by Perez et al. [26,27,28]. It was identified as belonging to the species *Clavispora lusitaniae* strain 146 (NCBI accession number KY442860). Yeast selection was based on in vivo bioprotective activity against *P. digitatum* according to our previous studies on lemons [27,28]. The yeast strain *Rhodosporidium paludigenum* Fell & Tallman NCYC 2663 (syn. CBS 6565, NRRL Y-12958) was isolated in 1996 from a mangrove swamp in Florida. *Saccharomyces cerevisiae* CEN.PK2-1C (MATa; his3D1; leu2-3_112; ura3-52; trp1-289; MAL2-8c; SUC2) was purchased from the European *Saccharomyces Cerevisiae* Archive For Functional Analysis collection (www.uni-frankfurt.de/fb15/mikro/euroscarf).

Yeast cultures were grown on YEPD medium containing yeast extract 5 g L^−1^, peptone 10 g L^−1^, dextrose 20 g L^−1^ (Merck, Darmstadt, Germany with pH set to 4.5 for routine use. The pathogenic strain of *P. digitatum* was provided by the Plant Pathology Lab from “Estación Experimental Obispo Colombres” (EEAOC, Tucumán, Argentina). “Eureka” lemons (*Citrus limon* (L.) Burm), grapefruits (*Citrus x paradisi*), mandarins (*Citrus reticulata*), ‘Jaffa’ sweet orange (*Citrus sinensis*), and tangerines (*Citrus x tangerina*) used in this study were freshly harvested fruits from local crops in Tucumán Province, Argentina. Harvested fruits did not receive any postharvest treatment and were used immediately or kept at 8 °C for no more than 4 days before use. Only grapefruits (*Citrus x paradisi*) were harvested 15 days before the assays and were coated with wax (carnauba).

### 2.2. Analysis of the Effect of Exposure Time and Yeast Concentration on Biocontrol Efficiency

The efficiency of different concentrations of strain *C. lusitaniae* 146 on the protection of wounds against *P. digitatum* on lemons was studied according to Perez et al. [27]. Fifteen fruits were used in each treatment and the experiment was repeated four times. Thus, eight different yeast concentrations were tested using sixty lemons for each concentration. Disinfected fruits were wounded in the equatorial zone and then dipped in each yeast suspension for 2 min. After 24 h of incubation at 25 °C and high humidity (about 90%), the wounded and yeast-treated fruits were dipped in 10 L of a spore suspension of *P. digitatum* (1 × 10^6^ spores mL^−1^) for 2 min using net bags and incubated for 5 days under the same conditions. Twenty fruits were taken as a control; they were wounded and dipped only in a fungal spore suspension.

Yeast and fungal spore suspensions were prepared according to Perez et al. [26]. The initial yeast suspension of *C. lusitaniae* 146 contained 3.06 × 10⁸ colony forming units (CFU) mL^−1^ (prepared with 100% yeast inoculum). Dilutions were prepared as described in the caption of Figure 1A. The effect of dipping time on the protection efficiency against *P. digitatum* was evaluated by varying dipping times in the yeast suspension to 15 s, 30 s, 1 min, 1.5 min, 2 min, and 2.5 min.

Protection efficiency was calculated as the percentage of healthy lemons after treatment:(1)$$Protection\;efficiency\;\left(\%\right)\;=\;\frac{number\;of\;healthy\;fruit}{total\;number\;of\;fruit}\;\times 100.$$

### 2.3. The Effect of Yeast Growth Phase on Biocontrol Efficiency

The biocontrol efficiency of *C. lusitaniae* 146 against *P. digitatum* on lemons according to the growth phase of the yeast was studied as described by Perez et al. [28] with some modifications. A yeast preinoculum was prepared in YEPD medium and incubated at 25 °C and 160 rpm for 24 h. Erlenmeyers containing 250 mL YEPD medium were inoculated with the yeast and incubated for 12, 24, 48, and 72 h under the same conditions as previously described. Colony forming units (CFU mL^−1^) were determined after each incubation time. Cells were collected by centrifugation and washed twice with sterile saline solution. Cell suspensions for the different growth times were used for in vivo biocontrol tests in lemons.

### 2.4. In Vivo Antagonist Activity Against *P. digitatum* Infecting Other Citrus Fruits

The biocontrol spectrum of *C. lusitaniae* 146 against *P. digitatum* was also tested on grapefruits, mandarins, sweet oranges, and tangerines. The experiment was carried out as previously described for lemons. All experiments were performed at 25 °C.

### 2.5. Patulin Degradation by *C. lusitaniae 146*

Patulin degradation experiments for the tested yeasts were carried out according to Zhu et al. [34] with slight modifications. Yeast cultures were grown in YEPD medium (pH 4.5) with shaking (180 rpm) at 25 °C for 24 h, harvested by centrifugation, and washed twice with sodium phosphate buffer (0.05 M, pH 5.8). The cells were resuspended in the same buffer and adjusted to the optical density of 1.0 at 600 nm, corresponding to about 10^6^ CFU mL^−1^. Yeast suspensions were divided equally into 10 sterile Erlenmeyer flasks. Half of the flasks were placed into a water bath at 85 °C for 20 min to inactivate the cells. The heat-inactivated cultures were cooled to room temperature and patulin was added to all flasks to a final concentration of 10 µg mL^−1^. The same volume of sodium phosphate buffer in five flasks with the same concentration of patulin served as controls. All samples were incubated with shaking (180 rpm) at 25 °C, and the patulin concentration in the supernatants was determined after 48 h. Each treatment consisted of five replicates.

### 2.6. Determination of Patulin Concentration by HPLC

Water was purified using an Arium Pro Ultrapure Water System (Sartorius, Goettingen, Germany). LC-MS grade methanol was purchased from Th. Geyer (Hoexter, Germany). Patulin standard was purchased from Acros Organics (Geel, Belgium). Samples for patulin analysis were prepared by diluting 20 µL of the supernatant of yeast cultures with 980 µL of methanol/water/acetic acid (30:70:0.3). A calibration curve was constructed using 11 concentrations of pure patulin from 3 to 1000 µg L^−1^. Blank samples were analyzed after every fifth sample, and a quality control standard (patulin at 250 µg L^−1^) was included after every 10th sample. HPLC-MS/MS analysis was carried out using an Agilent 1290 Infinity II HPLC system coupled with an Agilent 6460 triple quadrupole (Agilent Technologies, Waldbronn, Germany). A Polaris 3 C18-ether column of 100 × 2 mm with a 3 µm particle size (Agilent Technologies, Waldbronn, Germany) kept at 40 °C was loaded with 5 µL of samples and eluted with a gradient of methanol in water. Solvent A was water with 0.1% acetic acid, and solvent B was methanol with 0.1% acetic acid. The gradient was programmed as follows: 0 to 0.2 min, 5% B; 0.2 to 4 min, 5% to 50% B; 4 to 6 min, 50 to 98% B; 6 to 8 min, 98% B; 8 to 9 min, 98% to 5% B; 9 to 14 min, 5% B.

The limit of detection (LOD) was determined as a concentration corresponding to 3.9 times of the standard deviation of the blank. The limit of quantification (LOQ) was set as 3.3 times the LOD value. An LOD of 0.7 µg L^−1^ and an LOQ of 2.7 µg L^−1^ were obtained.

### 2.7. Sensorial Analysis by Paired Comparison Test

A sensory analysis test was carried out, focused on the fruit aroma, to determine the preference of a panel of assessors for lemons treated with *C. lusitaniae* 146 and untreated lemons. For this purpose, a paired preference test was chosen [35], where assessors received two different samples and were asked which one was preferred.

First, a group of lemons were treated with a cell suspension of *C. lusitaniae* for 2 min as described in Section 2.2 without wounding or dipping in a fungal spore suspension. Another group of lemons did not receive any treatment. Both groups were stored for 24 h at 25 °C before use. All fruits were similar in color, size, shape, and appearance. The sensory analysis was carried out by a panel of 100 non-trained assessors, with ages ranging from 18 to 60 years, including male and female participants. The purpose of the test was explained to the participants, and they provided informed consent. The assessors were recruited one by one to a room with adequate ventilation and lighting. Each assessor evaluated one sample of each group, previously randomized to avoid position bias, and presented in similar clear plastic containers. A questionnaire was given to the assessors, and they were asked to select one sample according to their preference based on the perceived aroma.

### 2.8. Statistical Analysis

The protection efficiency was analyzed by ANOVA, and the mean values were compared with Tukey’s test at the 5% significance level, except for patulin degradation, which was analyzed by an unpaired t-test with Welch’s correction [36]. InfoStat/L software (Córdoba, Argentina) [37] and GraphPadPrism v. 8.4.3 (San Diego, CA, USA) were used for statistical analysis. Box plots show the first and third quartile, and the whiskers extend to the 1.5-fold interquartile distance.

The results of the sensorial analysis were analyzed by a chi-square (χ^2^) test, where the proposed hypotheses were no preferences are found (H0) and a preference was found (H1). Degrees of freedom were equal to 1 and the confidence level was 95% (α = 0.05) [35].

## 3. Results

### 3.1. Effect of Exposure Time, Yeast Concentration, and Yeast Growth Phase on Biocontrol Efficiency

The effect of varying cell concentrations of *C. lusitaniae* 146 in protecting wounds in lemons against *P. digitatum* infections was examined. Starting from an initial concentration of 3.06 × 10⁸ CFU mL^−1^ (100%), different dilutions as low as 0.5% of the initial concentration were tested. The ability of strain 146 to prevent infection of wounds in lemons by *P. digitatum* remained high up to 2.5% of the initial yeast concentration while the levels of control varied between 83% and 98%. At 1% and 0.5% of the yeast initial concentration, protection efficacies with 67% and 53%, respectively, were significantly lower (Figure 1A).

The effect of the exposure time (the period of time that the fruits were dipped in the yeast suspension) was also examined in the range from 15 s to 2.5 min. No significant effect of exposure time on the protection efficiency was observed; the level of control ranged from 86% to 95% for all exposure times (Figure 1B).

The biocontrol efficiency of *C. lusitaniae* 146 at different yeast growth phases against *P. digitatum* on lemons can be seen in Figure 2. The lowest efficiency (67%) was observed with cultures grown for 12 h, where the concentration of cells was 5.60 × 10^8^ CFU mL^−1^. A steady and high degree of protection was reached with cultures grown for 24, 48, and 72 h with cell concentrations of 5.19 × 10^8^, 2.66 × 10^9^, and 1.33 × 10^10^ CFU mL^−1^, respectively.

### 3.2. Control of *P. digitatum* in Different Citrus Fruits by *C. lusitaniae 146*

In order to test the protective activity of *C. lusitaniae* against the rot-causing fungus *P. digitatum* on other citrus fruits, in vivo biocontrol assays were carried out on grapefruits, mandarins, oranges, and tangerines. *C. lusitaniae* 146 proved to be efficient against *P. digitatum* infection in all tested fruits (Figure 3A–D). The greatest protection was achieved on oranges (93%). For mandarins and tangerines, strain 146 achieved high protection values of 72% and 88%, respectively. In the case of grapefruits, the protection achieved by *C. lusitaniae* 146 was the weakest; nevertheless, it was still considerable (65% efficiency). The lower degree of protection for the latter fruit can likely be explained by the fact that the grapefruits used in this study were wax treated, rather than freshly harvested.

### 3.3. Degradation of Patulin by *C. lusitaniae 146*

To determine whether *C. lusitaniae* is able to degrade patulin, viable and heat-inactivated cells of *C. lusitaniae* 146 were incubated in buffer containing 10 µg mL^−1^ patulin for 48 h. Pure buffer served as a negative control, and *R. paludigenum* and *S. cerevisiae* were used as positive controls. After 2 days, the patulin concentration was reduced in all treatments as compared with the control, but living cells removed larger amounts of patulin than heat-inactivated cells (Figure 4). Furthermore, *C. lusitaniae* and *R. paludigenum* reduced the concentration of patulin more efficiently than *S. cerevisiae*.

### 3.4. Sensory Analysis of Lemon Fruits Treated with *C. lusitaniae 146*

With the aim of finding out whether treatment with the yeast *C. lusitaniae* influenced the perception of fruits by consumers, a sensory test was carried out on 100 panelists for treated and untreated lemons. According to the results, 58% of the assessors preferred treated lemons and 42% preferred the untreated ones (Appendix A). The data were analyzed by comparing tabulated values of the χ^2^ statistic for two-sided preference test with the calculated statistic (Appendix A). The test did not detect any preference; therefore, the observed differences were due to random chance alone.

## 4. Discussion

In our previous studies, we have demonstrated several advantages of the use of the killer yeast *C. lusitaniae* 146 to prevent postharvest fungal infection in lemons. Strain 146 showed a consistent preventive effect against green mold over a whole lemon harvest period, both at room and at low temperatures, proving to be a suitable protective agent for lemons for export to overseas markets. The strain also exhibited tolerance to commercial fungicides, which would potentially allow for a combined application of fungicides and the biocontrol agent [27]. *C. lusitaniae* 146 was highly efficient against both fungicide-sensitive and fungicide-resistant strains of *P. digitatum* [28]. Due to these promising results, in this work we studied additional conditions that can influence the performance of this yeast in biological control. The treatment conditions were optimized for the parameters yeast growth phase, cell density, and dipping time. Additionally, we have determined the efficiency of the control of *P. digitatum* on other citrus fruits, assessed the sensory impact of treatment on the aroma of fruits, and investigated the ability of strain 146 to degrade the mycotoxin patulin.

The results showed that reducing the concentration of *C. lusitaniae* 146 cells applied to wounded lemons up to 2.5% of the initial cell density did not affect the efficiency of the yeast against *P. digitatum*. This density is 2 orders of magnitude lower than the density necessary for efficient protection by some of the previously described biocontrol yeasts [38]. Even at the lowest tested concentration (0.5% of the initial inoculum), the yeast provided a 53% level of protection against fruit rot. Moreover, a 15 s-long dip of the fruit into a yeast suspension was sufficient to provide a 91% wound protection. These results indicate that if the yeast is used as a biocontrol agent in lemon packinghouses, the exposure of lemons to the yeast can be reduced to a minimum, thus allowing for rapid fruit processing on the packaging line. The results also indicate that the concentration of the yeast needed to prevent infection can be substantially reduced without loss of efficiency.

Yeast cells harvested after 12 h of growth had a rather low protection efficiency as compared with cells harvested after 24 h. The effect of the growth phase on the biocontrol efficiency could be due to the fact that killer toxin production was not sufficient at the logarithmic growth stage, based on the reports by Marquina et al. [39] and Buzzini and Martini [40] that toxin activity of different killer yeast strains reached a peak during the early stationary phase.

Ideally, a biocontrol agent should be effective against different postharvest pathogens in a wide range of fruits. Narrow-range activity is seen as a limitation for commercial success. Broad-range activity of biocontrol yeast appears to be associated with tolerance to stress caused by reactive oxygen species, extreme temperatures, and osmotic stress [41,42]. The biocontrol efficacy of *C. lusitaniae* 146 has so far been tested primarily in lemons. In the present work, the protective capability of strain 146 for mandarins, grapefruits, oranges, and tangerines was studied. Unlike other yeasts, *C. lusitaniae* proved to possess a wide range of protective activity within citrus, without a need for adaptation to stress factors or combination with additional products, such as Generally Recognized as Safe (GRAS) salts [43]. A broad spectrum of citrus fruits protected by the yeast is a valuable trait that expands the options for the applications of biocontrol agents based on strain 146. Furthermore, *C. lusitaniae* 146 may be able to control further *Penicillium* species beyond *P. digitatum*. Many *Penicillium* species produce patulin [11]; therefore, the ability of *C. lusitaniae* 146 to degrade patulin was investigated. *C. lusitaniae* degraded patulin as fast as *R. paludigenum* and much faster than *S. cerevisiae*. The efficiency of patulin degradation by *S. cerevisiae* varies among strains [44,45]. The *S. cerevisiae* strain used in this work degraded patulin with a substantially lower efficiency than *C. lusitaniae* and *R. paludigenum.* This strain was derived from a cross between North American strains [46,47], which were reported to degrade patulin with a comparatively low efficiency [44]. The degradation of patulin by both *C. lusitaniae* and *R. paludigenum* was very efficient (Figure 4), but *R. paludigenum* was also reported to stimulate patulin synthesis in *P. expansum*. Under certain conditions, patulin levels in fruits treated with *R. paludigenum* increased [29]. It is not known whether *C. lusitaniae* stimulates patulin synthesis in *P. expansum*. If not, *C. lusitaniae* may be a promising biocontrol agent against *P. expansum* and/or other patulin-producing *Penicillium* pathogens. We suggest that patulin degradation as well as stimulation of patulin synthesis be included in the assessment of biocontrol agents for fruit pathogens producing patulin such as *P. expansum*, *Penicillium griseofulvum*, *Penicillium polonicum*, *Penicillium brevicompactum*, *Penicillium crustosum*, and *Penicillium cyclopium* [11].

Interestingly, heat-inactivated cells of all three yeast species reduced patulin concentration as well. The reduction was statistically significant but much lower than the reduction caused by living cells, especially for *C. lusitaniae* and *R. paludigenum.* We assume that the removal of patulin from solution by heat-inactivated cells was caused by adsorption because many microorganisms [48,49], including yeast [50,51], reduce patulin concentration in solution by physical adsorption. The process is likely to be reversible, which further diminishes its attractiveness for applications.

It is expected that postharvest biological control can improve the safety of fruits, in addition to extending their shelf life. Treatment with biocontrol agents should, however, not impair the quality and sensory properties of fruits. During storage of treated products, protective cultures are expected to grow and release enzymes and metabolites that could affect the properties of food [52]. Yeasts are known to produce a wide range of volatile compounds, including metabolites with pungent aroma [30]. Our results suggest that the killer yeast *C. lusitaniae* 146 does not negatively affect the aroma perception of fruits by consumers, as no preference regarding fruit aroma was found when assessors compared untreated and yeast-treated lemons in a sensory analysis. As far as we know, this is the first report of the impact of a biocontrol yeast on the sensory properties of citrus fruits.

## 5. Conclusions

The study demonstrated several advantages of the killer yeast *C. lusitaniae* as a biological control agent. Efficient protection was achieved after short exposure to a low concentration of yeast cells. Protection of different citrus fruits and the ability to degrade patulin corroborated the assessment of *C. lusitaniae* as a wide-range biocontrol agent. The yeast treatment has not affected fruit aroma; therefore, *C. lusitaniae* is a promising organism for the formulation of an effective biocontrol agent for citrus fruits.

## Figures and Tables

**Figure 1 microorganisms-08-01477-f001:**
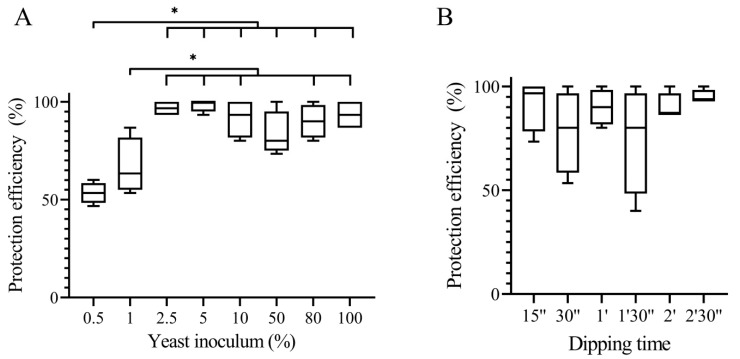
Efficiency of *Clavispora lusitaniae* 146 to protect lemon wounds against *Penicillium digitatum* according to the yeast concentration (**A**) and dipping time (**B**). (**A**) The values on the horizontal axis indicate the percentage of the dilution prepared from an initial concentration (100%) of 3.06 × 10^8^ colony forming units (CFU) mL^−1^. (**B**) Dipping times from 15 s to 2.5 min were tested. Significant differences in the efficiency are indicated by asterisks according to Tukey’s test (* indicates *p* < 0.05, *n* = 4).

**Figure 2 microorganisms-08-01477-f002:**
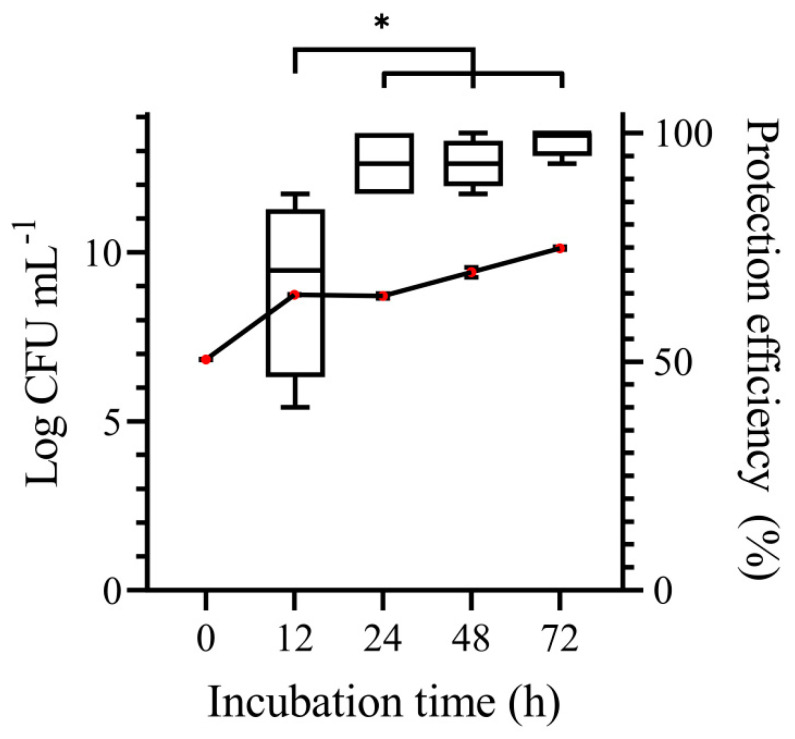
Biocontrol efficiency of *C. lusitaniae* 146 against *P. digitatum* in lemons according to yeast growth phase. Box plots represent the efficiency of protection against *P. digitatum*. Line graph indicates the cell density of *C. lusitaniae* 146. Significant differences in the efficiency are indicated by asterisks according to Tukey’s test (* indicates *p* < 0.05, *n* = 4).

**Figure 3 microorganisms-08-01477-f003:**
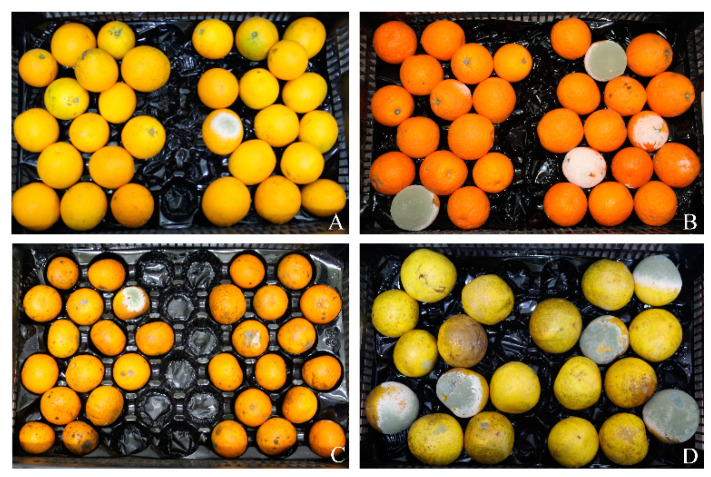
Efficiency of *C. lusitaniae* 146 in the protection of wounds against *P. digitatum* on four citrus varieties. The upper panel shows in vivo results on the control of green mold by *C. lusitaniae* 146 after 5 days of incubation at 25 °C. From top left to bottom right: oranges (**A**), tangerines (**B**), mandarins (**C**), and grapefruits (**D**). The bottom panel shows the protection efficiency graphically (*n* = 4).

**Figure 4 microorganisms-08-01477-f004:**
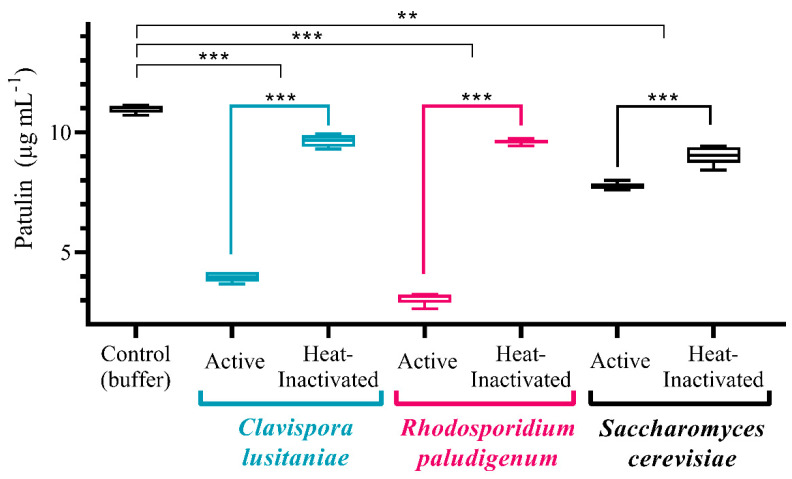
Removal of patulin from solution by *C. lusitaniae, Rhodosporidium paludigenum*, and *Saccharomyces cerevisiae*. Cells of the yeasts, native and heat-inactivated, were incubated in phosphate buffer with patulin for 48 h and the patulin concentration in the supernatant was determined. The significance of differences between means was determined by an unpaired t-test with Welch’s correction (*n* = 5) with *** indicating *p* < 0.0001 and ** indicating *p* < 0.001.

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
