# Peer review of "Protection of Citrus Fruits from Postharvest Infection with *Penicillium digitatum* and Degradation of Patulin by Biocontrol Yeast *Clavispora lusitaniae* 146"

_microorganisms, 2020, doi:10.3390/microorganisms8101477_

Round 1
Reviewer 1 Report
The authors assessed the effectiveness of a yeast as a biocontrol agent against Penicillium digitatum on citrus fruits, finding that a low cell concentration and short dipping times were still effective, and did not affect sensory properties. The work is interesting and practical. The manuscript is well-written; few minor typos were noticed.
My only questions are (1) if use of human participants was approved by an ethics review panel and (2) if participants provided informed consent.
Reviewer 2 Report
The paper authored by Diaz et al. entitled “Protection of citrus from postharvest infection with Penicillium digitatum and degradation of patulin by biocontrol yeast Clavispora lusitaniae 146” reports the effect of yeast concentration, yeast growth phase as well as dipping time on biocontrol activity for the protection of Citrus varieties against the fungal pathogen Penicillium digitatum, a major pathogen of Citrus during postharvest stage. The authors also studied the patulin mycotoxin degradation by the biocontrol yeast strain. Finally, they carried out a sensorial analysis to check if the yeast treatment has an impact on aroma perception.
The paper is well written and is presenting original approach to identify the best biocontrol strain to apply on postharvest Citrus varieties.
To my opinion, it can be accepted for publication after minor revisions
- the deletion of fig. 5. This part does not need visual representation as there are only two values.
- L146-148 : Explanation about LOD and LOQ targeted values
- L209 : did the authors try protection using wax treated fruits for other varieties to corroborate the hypothesis? Should be discussed a little bit further.
